# Association between surgeon grade and implant survival following hip and knee replacement: a systematic review and meta-analysis

Timothy J Fowler ,[1] Alex L Aquilina,[1] Ashley W Blom,[1,2] Adrian Sayers ,[1] Michael R Whitehouse [1,2]

AS and MRW are joint senior authors.

¹Musculoskeletal Research Unit, Learning and Research Building, Southmead Hospital, University of Bristol Medical School, Bristol, UK
²National Institute for Health Research Bristol Biomedical Research Centre, University Hospitals Bristol NHS Foundation Trust, University of Bristol, National Institute for Health Research, Bristol, UK

**Correspondence to**
Timothy J Fowler;
t.j.fowler@bristol.ac.uk

## ABSTRACT

**Objective** To investigate the association between surgeon grade (trainee vs consultant) and implant survival following primary hip and knee replacement.

**Design** A systematic review and meta-analysis of observational studies.

**Data sources** MEDLINE and Embase from inception to 6 October 2021.

**Setting** Units performing primary hip and/or knee replacements since 1990.

**Participants** Adult patients undergoing either a primary hip or knee replacement, predominantly for osteoarthritis.

**Intervention** Whether the surgeon recorded as performing the procedure was a trainee or not.

**Primary and secondary outcome measures** The primary outcome was net implant survival reported as a Kaplan-Meier survival estimate. The secondary outcome was crude revision rate. Both outcomes were reported according to surgeon grade.

**Results** Nine cohort studies capturing 4066 total hip replacements (THRs), 936 total knee replacements (TKRs) and 1357 unicompartmental knee replacements (UKRs) were included (5 THR studies, 2 TKR studies and 2 UKR studies). The pooled net implant survival estimates for THRs at 5 years were 97.9% (95% CI 96.6% to 99.2%) for trainees and 98.1% (95% CI 97.1% to 99.2%) for consultants. The relative risk of revision of THRs at 5 and 10 years was 0.88 (95% CI 0.46 to 1.70) and 0.68 (95% CI 0.37 to 1.26), respectively. For TKRs, the net implant survival estimates at 10 years were 96.2% (95% CI 94.0% to 98.4%) for trainees and 95.1% (95% CI 93.0% to 97.2%) for consultants. We report a narrative summary of UKR outcomes.

**Conclusions** There is no strong evidence in the existing literature that trainee surgeons have worse outcomes compared with consultants, in terms of the net survival or crude revision rate of hip and knee replacements at 5–10 years follow-up. These findings are limited by the quality of the existing published data and are applicable to countries with established orthopaedic training programmes.

**PROSPERO registration number** CRD42019150494.

## INTRODUCTION

Hip and knee replacements are effective surgical interventions for the treatment of

### Strengths and limitations of this study

► To our knowledge, this is the first meta-analysis of the association between surgeon grade and implant survival following hip and knee replacement.
► We performed a comprehensive systematic review according to current best practice guidelines.
► The findings of this review are limited by the strength of the existing published data from a relatively small number of predominantly retrospective observational studies.

end stage degenerative conditions of the hip and knee.[1 2] More than 200 000 are performed per year in the UK alone.[3] These procedures are performed by surgeons at various stages in their training, with varying levels of senior supervision. Contemporary training practices must ensure a balance between protecting development opportunities for the next generation of surgeons, while limiting the exposure of patients to unnecessary risk during the training process.

Implant survival, which is determined by the absence of revision surgery, is an important and commonly used measure of surgical performance.[4 5] Net survival estimates are calculated using statistical methods of survival analysis (eg, Kaplan-Meier analysis), which look at time to a defined failure 'event' (eg, revision) and account for censored data that arise due to incomplete follow-up or death.[6] Another commonly reported metric is crude revision rate, which is defined as the observed number of failure events in a specified period of time.

The survival of hip and knee replacements according to surgeon grade is poorly understood. Higher rates of complications and longer operative times have been identified in orthopaedic procedures performed by trainees.[7 8] Radiographic studies comparing

trainee and consultant joint replacement have identified differences in acetabular anteversion,[9] hip centre of rotation[10] and various measures of knee replacement component positioning.[11] However, the relative impact of these findings on implant survival has not been established. It has been suggested that when trainees are appropriately supervised, they can obtain comparable functional outcomes and implant survivorship to their consultant colleagues when performing total hip replacement (THR),[12–14] total knee replacement (TKR)[15] and unicompartmental knee replacement (UKR).[16]

The aim of this study was to conduct a systematic review and meta-analysis using the existing literature on the association between surgeon grade (trainee vs consultant) and implant survival outcomes in hip and knee replacement surgery. We aimed to answer the question— do trainees achieve equivalent implant survival outcomes to consultants when performing primary hip and knee replacement?

## METHODS
This review was conducted using methods described in the Cochrane Handbook for Systematic Reviews of Interventions, with reporting in accordance with the Meta-analyses Of Observational Studies in Epidemiology checklist.[17 18] The study was registered with the PROSPERO database at inception (CRD42019150494).

### Data sources and search strategy
We searched for cohort studies reporting implant survival estimates and/or revision rates of hip or knee replacements, according to surgeon grade. Separate searches were performed for hips and knees. We conducted searches of MEDLINE and Embase from inception to 6 October 2021. Searches used keywords and Medical Subject Headings terms relating to hip and knee replacement, implant survival, revision surgery and surgeon grade (see online supplemental methods). There were no language restrictions. Titles and abstracts of potentially relevant non-English language citations were translated. We manually screened the bibliographies of full text articles and used Web of Science citation tracking to identify additional relevant studies.

### Eligibility criteria
We included studies if they involved predominantly unselected adult patients (≥18 years old) undergoing primary hip or knee replacement (including THR, TKR, UKR and hip resurfacing), predominantly for the treatment of osteoarthritis. Included articles needed to report the primary and/or secondary outcome measure for two different groups of surgeons defined according to their grade (eg, trainee vs consultant). We defined a minimum follow-up of 5 years and articles that did not clearly define the length of follow-up were excluded. For example, we excluded studies reporting the revision rate 'per 100 component years', as these did not explicitly define the

length of follow-up. We excluded studies in which the index operation was performed prior to 1990; thereby, including studies that are representative of contemporary training practices, but also allowing for inclusion of studies reporting in excess of 30 years of follow-up (see online supplemental methods).

### Primary exposure
The primary exposure was whether the surgeon recorded as performing the procedure was a trainee or not. Surgeon grade is a measure of the designated level of surgical experience and seniority, which we considered to be a binary variable: either 'trainee' or 'consultant'. Consultant surgeons have completed their formal training in orthopaedic surgery and have been appointed to a senior position in which they can practice independently and supervise trainee surgeons. The term 'consultant' is used synonymously with 'attending surgeon' in many healthcare settings including the USA. Additional terms used to describe this variable were deemed eligible during screening (eg, Trainee: registrar; resident; junior/young surgeon; fellow. Consultant: attending; senior surgeon; trainer).

### Outcome measures
The primary outcome was net implant survival, reported as a Kaplan-Meier survival estimate. The secondary outcome measure was crude revision rate, which was defined as the observed number of revisions in a specified period of time.

### Screening and data extraction
Two authors (TJF and ALA) independently screened all titles and abstracts of journal articles using Rayyan (Rayyan QCRI, Doha).[19] Studies were initially screened for relevance according to information contained within the title and abstract. Cases of disagreement were resolved through rereview and consensus. Full texts of potentially relevant studies were reviewed in detail and disagreements on final inclusion were resolved through discussion with a senior author (MRW). Specific indications for exclusion were documented following full-text review (figure 1 and online supplemental methods).

Data were extracted in duplicate using a standardised proforma. We recorded data on the following: healthcare setting, study period, implant type, age, sex, indication, level of supervision, crude revision rate and net implant survival estimates (including CIs)). Life tables were reviewed, and estimates were extracted for all available 5-year intervals of follow-up. Discrepancies in data collection were resolved through rereview and consensus. Where survival estimates, CIs and revision rates were incompletely reported, we contacted corresponding authors to request missing data.

### Risk of bias and quality of evidence assessment
The risk of bias was assessed using the Cochrane Risk of Bias in Non-randomised Studies - of Interventions (ROBINS-I) tool for the risk of bias in non-randomised

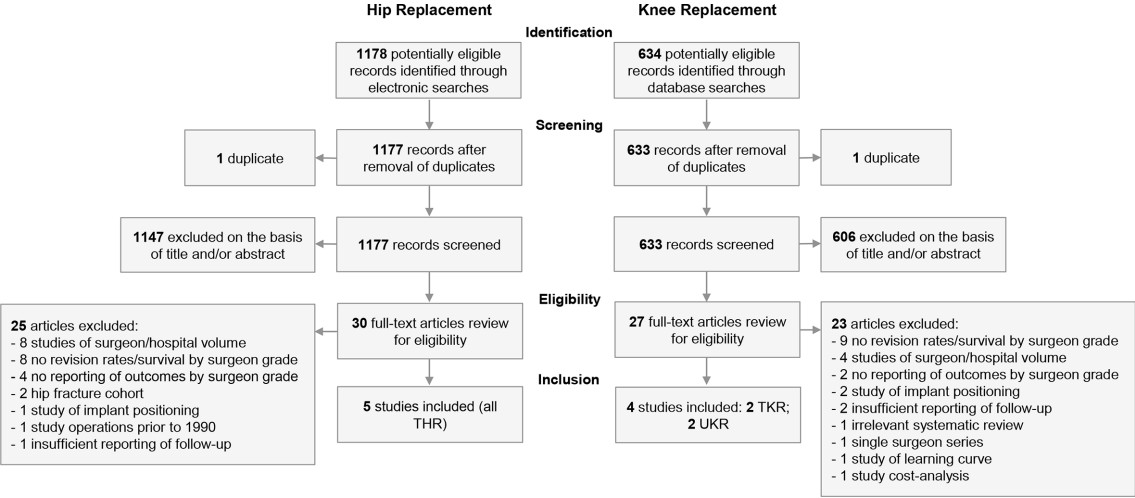

**Figure 1** Study flow diagram. THR, total hip replacement; TKR, total knee replacement; UKR, unicompartmental knee replacement.

cohort studies.[20] We assessed the quality of evidence for each outcome using the Grading of Recommendations Assessment, Development and Evaluation (GRADE) approach, which considers the imprecision, inconsistency, indirectness and risk of bias in a body of evidence.[21]

### Statistical analysis
Statistical analysis was performed using Stata (V.SE 15.1; StataCorp). For the primary outcome measure of net implant survival, we performed separate meta-analyses for each implant type, by surgeon grade and length of follow-up. We pooled survival estimates, assuming that survivorship approximated risk, with fixed effects meta-analysis weighting each study on the overall pooled estimate according to its SE, which was calculated from published CIs; an established method for the meta-analysis of implant survival estimates described by Evans *et al*.[4 5] The effect size (survival) for trainees and consultants, was compared using a Wald test. For the secondary outcome measure, we derived and meta-analysed the relative risk (RR) of revision for each implant type by surgeon grade and length of follow-up. We used a fixed effects model using the Mantel-Haenszel method.[22] Heterogeneity was assessed with chi-squared tests, with $I^2$ used to quantify inconsistency.[23] Publication bias was assessed by inspecting funnel plot symmetry.[24]

### Patient and public involvement
There was no direct patient or public involvement in the design or conduct of this review.

### RESULTS
Separate searches for hip and knee replacements identified 1178 and 634 articles, respectively. After removal of duplicates and abstract screening, 30 hip papers and 27 knee papers remained. Through review of full text articles, we identified five hip and four knee studies eligible for inclusion. This process of review is summarised

as a flow diagram in figure 1 and the characteristics of included studies are summarised in table 1. Six studies were conducted in the UK, with the remaining three studies originating from France, Switzerland and Japan.

### Risk of bias assessment
Online supplemental table 1 provides a summary of the ROBINS-I assessment, which indicates a moderate to severe risk of bias in all studies. Funnel plot asymmetry and statistical tests for funnel plot asymmetry as a means of assessing publication bias were not applicable due to the small number of studies.[25]

### Hip replacement
The five included hip studies represent 1464 THRs performed by trainees and 2602 THRs performed by consultants, with follow-up ranging from 5 to 10 years. Four studies were retrospective cohort studies[12 13 26 27]; one was a non-randomised prospective cohort study.[28] No articles on hip resurfacing met the inclusion criteria. One author provided additional unpublished data in the form of net survival estimates.[27] Reidy *et al* reported survival estimates, but no CIs.[13] Net survival estimates and corresponding CIs were thus extracted from three studies at 5 years and one study at 10 years. Crude revision rates were reported in three studies at 5 years and two studies at 10 years.

### Primary outcome: net implant survival (THR)
Meta-analysis showed net survivorship of 97.9% (95% CI 96.6% to 99.2%) at 5 years for THRs performed by trainees, compared with 98.1% (95% CI 97.1% to 99.2%) for THRs performed by consultants (figure 2). There was no strong evidence of an association between surgeon grade and net implant survival at this interval of follow-up (Wald test: p=0.74).

Meta-analysis was not possible for the 10-year data given the availability of only one study for this time point. In a cohort of 1082 reverse hybrid THRs, Jain *et al*

**Table 1** Characteristics of included studies

| Source, year | Country | Study period | Study design | Implant | Surgeon grade terminology (primary exposure) | Follow-up (years) | No of cases (trainee) | Implant brand (stem/cup if hip) | Sex (% female) | Mean age (SD or range) | Indication (% OA) | Supervision reported | Survival analysis | Revision rates reported | ROBINS-I overall risk of bias* |
|---|---|---|---|---|---|---|---|---|---|---|---|---|---|---|---|
| Hasegawa,[28] 2015 | Japan | 2006–2010 | PC | THR | Trainee vs instructor | 5 | 483 (259) | Multiple | – | 61.3 (SD 11.6) | – | No | Yes | No | Serious |
| Jain,[27] 2018 | UK | 2005–2012 | RC | THR | Trainee vs consultant | 5, 10 | 1082 (348) | Corail/multiple | 61.3 | 69.2 (21–94) | 91.0 | No | Yes (Add.) | Yes | Moderate |
| Müller,[26] 2013 | Switzerland | 2005–2006 | RC | THR | Junior vs senior | 5 | 130 (43) | Quadra-H / Versafit-CC | 52.0 | 64 (SD 12.36) | 86.0 | No | Yes | Yes | Serious |
| Palan,[12] 2009 | UK | 1999–2002 | RC | THR | Trainee vs consultant trainer | 5 | 1501 (528) | Exeter/multiple | – | 68.4 (21–94) | – | No | No | Yes | Moderate |
| Reidy,[13] 2016 | UK | 2003–2004 | RC | THR | Trainee vs consultant | 10 | 870 (286) | Multiple | 60.5 | 69.5 (37–94) | 94.8 | Yes | Yes (no CIs) | Yes | Moderate |
| Faulkner,[15] 2017 | UK | 2003–2004 | RC | TKR | Trainee vs consultant | 5, 10 | 686 (236) | Multiple | – | 69.9 (30–94) | 93.1 | No | Yes (Add.) | Yes | Moderate |
| Hernigou,[29] 2009 | France | 1990–1995 | RC | TKR | Young (<30) vs senior | 10, 15 | 250 (150) | Ceraver Hermes | 69.7 | 73 (46–88) | – | No | Yes | No | Serious |
| Bottomley,[16] 2016 | UK | 1998–2008 | RC | UKR | Trainee vs consultant | 10 | 1084 (673) | Oxford | 51.4 | 66.5 (SD 9.6) | 100 | Yes | Yes | Yes | Moderate |
| Alvand,[30] 2021 | UK | 2009–2015 | RC | UKR | Trainee vs consultant | 5 | 273 (118) | Oxford | 49.5 | 67.8 (SD 10.1) | 98.2 | Yes | No | Yes | Moderate |

*See online supplemental table 1 for full risk of bias assessment.
Add., additional data provided by author; OA, osteoarthritis; PC, prospective cohort; RC, retrospective cohort; ROBINS-I, Risk of Bias in Non-randomised Studies - of Interventions; THR, total hip replacement; TKR, total knee replacement; UKR, unicompartmental knee replacement.

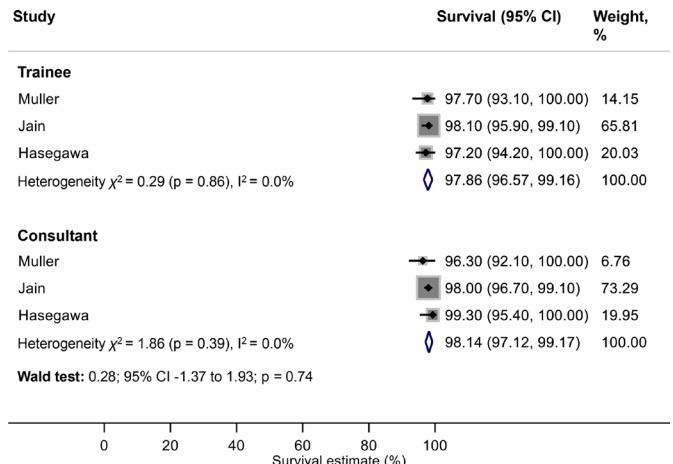

| Study | | Survival (95% CI) | Weight, % |
|---|---|---|---|
| **Trainee** | | | |
| Muller | | 97.70 (93.10, 100.00) | 14.15 |
| Jain | | 98.10 (95.90, 99.10) | 65.81 |
| Hasegawa | | 97.20 (94.20, 100.00) | 20.03 |
| Heterogeneity $\chi^2$ = 0.29 (p = 0.86), $I^2$ = 0.0% | | 97.86 (96.57, 99.16) | 100.00 |
| **Consultant** | | | |
| Muller | | 96.30 (92.10, 100.00) | 6.76 |
| Jain | | 98.00 (96.70, 99.10) | 73.29 |
| Hasegawa | | 99.30 (95.40, 100.00) | 19.95 |
| Heterogeneity $\chi^2$ = 1.86 (p = 0.39), $I^2$ = 0.0% | | 98.14 (97.12, 99.17) | 100.00 |

**Wald test:** 0.28; 95% CI -1.37 to 1.93; p = 0.74

**Figure 2** Meta-analysis of net implant survival of THRs at 5 years according to surgeon grade. THRs, total hip replacements.

demonstrated overall 97.2% implant survival at 10 years. Additional data provided by the author indicate that they found no evidence of a difference in implant survival according to surgeon grade (Trainee: 98.1%; 95% CI 95.9 to 99.1; Consultant: 96.7%; 95% CI 94.7 to 97.9).[27]

### Secondary outcome: crude revision rate (THR)

Meta-analysis showed no strong evidence of an association between surgeon grade and the crude revision rate at 5 or 10 years. The RR of revision at 5 and 10 years was 0.88 (95% CI 0.46 to 1.70) and 0.68 (95% CI 0.37 to 1.26), respectively (figure 3).

### Knee replacement

The four knee studies represent 1177 knee replacements (TKR n=386; UKR n=791) performed by trainees and 1116 knee replacements (TKR n=550; UKR n=566) performed by consultants, with follow-up ranging from 5 to 15 years. All four were retrospective cohort studies.[15 16 29 30] Two studies reported on TKRs,[15 29] and two studies reported on UKRs.[16 30]

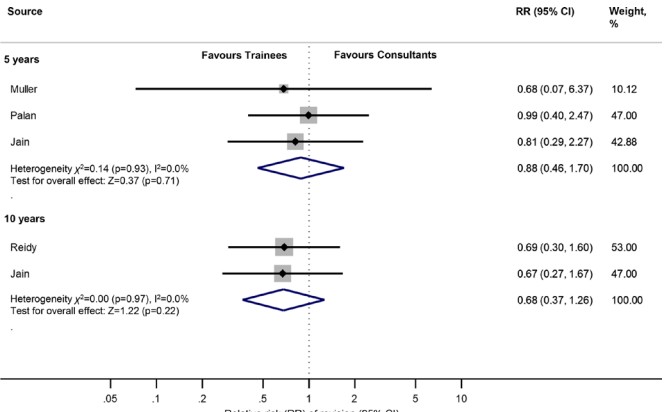

| Source | | RR (95% CI) | Weight, % |
|---|---|---|---|
| **5 years** | Favours Trainees — Favours Consultants | | |
| Muller | | 0.68 (0.07, 6.37) | 10.12 |
| Palan | | 0.99 (0.40, 2.47) | 47.00 |
| Jain | | 0.81 (0.29, 2.27) | 42.88 |
| Heterogeneity $\chi^2$=0.14 (p=0.93), $I^2$=0.0%<br>Test for overall effect: Z=0.37 (p=0.71) | | 0.88 (0.46, 1.70) | 100.00 |
| **10 years** | | | |
| Reidy | | 0.69 (0.30, 1.60) | 53.00 |
| Jain | | 0.67 (0.27, 1.67) | 47.00 |
| Heterogeneity $\chi^2$=0.00 (p=0.97), $I^2$=0.0%<br>Test for overall effect: Z=1.22 (p=0.22) | | 0.68 (0.37, 1.26) | 100.00 |

**Figure 3** Meta-analysis of the relative risk of revision of THRs at 5 and 10 years according to surgeon grade. THRs, total hip replacements.

| Source | | Survival (95% CI) | Weight, % |
|---|---|---|---|
| **Trainee** | | | |
| Hernigou | | 96.00 (93.00, 100.00) | 38.17 |
| Faulkner | | 96.30 (92.60, 98.10) | 61.83 |
| Heterogeneity $\chi^2$=0.02 (p=0.90), $I^2$=0.0% | | 96.19 (94.02, 98.35) | 100.00 |
| **Consultant** | | | |
| Hernigou | | 96.00 (93.00, 100.00) | 36.44 |
| Faulkner | | 94.60 (91.10, 96.40) | 63.56 |
| Heterogeneity $\chi^2$=0.39 (p=0.53), $I^2$=0.0% | | 95.11 (93.00, 97.22) | 100.00 |

**Wald test:** 1.08; 95% CI -1.95 to 4.10; p=0.49

**Figure 4** Meta-analysis of net implant survival of TKRs at 10 years according to surgeon grade. TKRs, total knee replacements.

With regard to the two TKR studies, Faulkner *et al* provided additional unpublished survival data from which we calculated corresponding CIs for their published survival estimates.[15] Net survival estimates and CIs were thus extracted from both TKR studies at 10 years, which permitted meta-analysis of this primary outcome measure. Crude revision rates were only available from one TKR study at each 5-year interval of follow-up.

With regard to the two UKR papers, net survival estimates were only available from one study.[16] Crude revision rates were available from one study at 5 years and one study at 10 years.[16 30] Meta-analysis was not feasible, thus we provide a narrative summary of UKR outcomes.

### Primary outcome: net implant survival (TKR)

Meta-analysis showed net survivorship of 96.2% (95% CI 94.0% to 98.4%) at 10 years for TKRs performed by trainees, compared with 95.1% (95% CI 93.0% to 97.2%) for TKRs performed by consultants (figure 4). There was no strong evidence of an association between surgeon grade and net implant survival at this interval of follow-up (Wald test: p=0.49).

### Secondary outcome: crude revision rate (TKR)

Two studies reported crude revision rates according to surgeon grade; however, with data from only one study available at each interval of follow-up, meta-analysis was not feasible. Instead, we provide a narrative summary. Faulkner *et al* provided additional unpublished data, which indicated crude revision rates at 5 years for trainees and consultants of 2.1% and 4.4%, respectively.[15] This rises to 3.4% (trainees) and 5.8% (consultants) at 10 years. These data represent a RR of revision of 0.49 (95% CI 0.19 to 1.28) at 5 years and 0.60 (95% CI 0.28 to 1.31) at 10 years. Hernigou published crude revision rates at 15 years of 2.7% for junior surgeons and 4.0% for senior surgeons, which represents a RR of revision of 0.68 (95% CI 0.17 to 2.64).[29]

### Unicompartmental knee replacement

Both UKR studies were conducted in the same centre but capture separate cohorts of patients.[16 30] Bottomley *et al* conducted a retrospective cohort study of 1084 consecutive UKRs performed between 1998 and 2008. They demonstrated that consultants and trainees had cumulative 9-year survival estimates of 93.9% and 93.0%, respectively. They found no strong evidence of a difference in implant survival between the groups (log rank: p=0.30).[16] These data represent crude revision rates at 10 years of 4.6% and 3.6% for trainees and consultants, respectively (RR 1.26; 95% CI 0.69 to 2.31). Trainees were supervised by a scrubbed consultant in 48% of cases.

Alvand *et al* reported a series of 273 UKRs performed between 2009 and 2015. They did not report net survival estimates according to surgeon grade. However, they reported crude revision rates at 5 years of 0.8% and 2.6% for trainees and consultants, respectively. These data represent a RR of revision of 0.33 (95% CI 0.04 to 2.90). Trainees were supervised by a scrubbed consultant in 100% of cases.

### Assessment of the quality of evidence

The GRADE assessment of the quality of evidence for each outcome indicates a low, or very low quality of evidence for all outcomes (table 2).

### DISCUSSION

The results of this study suggest that, in the context of contemporary practice, trainees do not achieve worse hip and knee replacement survival outcomes compared with their consultant colleagues at 5–10 years follow-up. We found no strong evidence of an association between surgeon grade and the net survival of THRs at 5 years (trainees: 97.9% vs consultants: 98.1%). There was no association between surgeon grade and the crude revision rate of THRs at either 5, or 10 years follow-up. Furthermore, we found no strong evidence of an association between surgeon grade and the net survival of TKRs at 10 years (trainees: 96.2% vs consultants: 95.1%). Our narrative summary of two studies, highlights that there is no evidence in the existing literature of an association between trainee performed UKR and an increased risk of revision.

### Strengths and limitations

This review has a number of strengths. We conducted a comprehensive systematic review with an exhaustive search according to current best practice guidelines and published the protocol for the methodology at inception. However, the data captured by this review have several limitations, which we have attempted to address through quality of evidence assessment and risk of bias analysis. The GRADE assessment indicates a low to very low quality of evidence for each outcome. Furthermore, the ROBINS-I assessment indicates a moderate to severe risk of bias in the included studies. These findings are generally consistent with the predominantly retrospective design of the included studies. The conclusions of this review are therefore limited by the strength and quality of the existing published data, which originate from a relatively small number of observational studies.

Meta-analysis of outcome measures was only possible at 5 and 10 years for THRs and 10 years for TKRs, which limits the generalisability of our findings to these short and medium-term intervals of follow-up. Therefore, this review does not capture any differences in early failure rates that might exist between trainee and consultant cohorts before 5 years. The included studies originated from the UK, France, Switzerland and Japan, which limits the generalisability of the findings to countries with established orthopaedic training programmes.

Formal orthopaedic training is a long process (lasting up to 10 years in some countries); therefore, individual trainees have varying levels of experience, which are not captured by the binary variables used in this study, or in the existing literature. The included studies did not provide sufficient data to perform meaningful adjustment or sensitivity analysis according to specific training grade, or the level of senior supervision. Furthermore, our study captures cases performed between 1990 and 2015 (table 1) and we were unable to adjust for variations in training practices (such as the level of senior supervision) that may have occurred over this 25-year period.

Implant survival is a key determinant of good outcome in joint replacement surgery and is the sole variable considered in the current benchmarking strategies for the assessment of implant components. However, this review did not consider other factors that may be important when evaluating surgical outcomes, such as patient reported outcome measures, or complications other than failure, which have previously been found to occur in higher rates when joint replacements are performed by less experienced surgeons.[7 8]

Published literature did not consistently report age, sex, comorbidities, implant design or the level of senior supervision; making it very difficult to adjust for these variables. Methods of categorising the procedural complexity of a hip or knee replacement are not widely used in the orthopaedic literature and were not reported by any of the studies included in this review. Therefore, it was not possible to adjust for this factor. It is reasonable to suggest that the predominantly superior survival outcomes observed in the trainee cohorts are a product of patient selection and close senior supervision, with good trainers selecting appropriately complex cases for their trainees.

### Comparison with other studies

A single study was excluded because the THRs under follow-up were performed prior to 1990[31]; thus not considered representative of contemporary training practices. The authors of this 10-year study of 413 THRs reported a significantly higher rate of revision for trainees, with 15 of 16 revised hips performed by trainees. Inclusion of this study in our meta-analysis of 10-year THR crude revision

**Table 2** GRADE summary of findings table

| Outcome | Follow-up (years) | Trainee revision/ cases,* n | Consultant revisions/ cases,* n | Net survival/relative risk (95% CI) | Participants (studies), n | Quality of evidence | Comments |
|---|---|---|---|---|---|---|---|
| THR: net implant survival | 5 | 650 | 1045 | NS: Trainee 97.9% (96.6% to 99.2%) NS: Consultant 98.1% (97.1% to 99.2%) | 1695 (3)[26–28] | Very low | Serious ROB, indirectness and imprecision |
| | 10 | 348 | 734 | NS: Trainee 98.1% (95.9% to 99.1%) NS: Consultant 96.7% (94.7% to 97.9%) | 1082 (1)[27] | Low | Serious indirectness and imprecision |
| THR: crude revision rate | 5 | 13/919 | 29/1794 | RR: 0.88 (0.46 to 1.70) | 2713 (3)[12 26 27] | Very low | Serious ROB, indirectness, and imprecision |
| | 10 | 13/634 | 40/1318 | RR: 0.68 (0.37 to 1.26) | 1952 (2)[13 27] | Low | Serious indirectness and imprecision |
| TKR: net implant survival | 5 | 236 | 450 | NS: Trainee 97.9% (95.0% to 99.2%) NS: Consultant 95.4% (93.0% to 97.0%) | 686 (1)[15] | Low | Serious imprecision |
| | 10 | 386 | 550 | NS: Trainee 96.2% (94.0% to 98.4%) NS: Consultant 95.1% (93.0% to 97.2%) | 936 (2)[15 29] | Very low | Serious inconsistency and imprecision |
| | 15 | 150 | 100 | NS: Trainee 91.0% (85.0% to 97.0%) NS: Consultant 92.0% (90.0% to 94.0%) | 250 (1)[29] | Very low | Serious inconsistency and very serious imprecision |
| TKR: crude revision rate | 5 | 5/236 | 20/450 | RR: 0.47 (0.18 to 1.25) | 686 (1)[15] | Low | Serious imprecision |
| | 10 | 8/236 | 26/450 | RR: 0.58 (0.27 to 1.27) | 686 (1)[15] | Low | Serious imprecision |
| | 15 | 4/150 | 4/100 | RR: 0.67 (0.17 to 2.60) | 250 (1)[29] | Very low | Serious inconsistency and very serious imprecision |
| UKR: net implant survival | 10 | 673 | 411 | NS: Trainee 93.0% (90.3% to 95.7%) NS: Consultant 93.9% (90.2% to 97.6%) | 1084 (1)[16] | Low | Serious imprecision |
| UKR: crude revision rate | 5 | 1/118 | 4/155 | RR: 0.33 (0.04 to 2.90) | 273 (1)[30] | Low | Serious imprecision |
| | 10 | 31/673 | 15/411 | RR: 1.26 (0.69 to 2.31) | 1084 (1)[16] | Low | Serious imprecision |

*Number of revisions not reported for net implant survival.
GRADE, Grading of Recommendations Assessment, Development and Evaluation; NS, net survival; ROB, risk of bias; RR, relative risk; THR, total hip replacement; TKR, total knee replacement; UKR, unicompartmental knee replacement.

rates increases the RR of revision to 1.12 (95% CI 0.66 to 1.92), in favour of THRs performed by consultants. One explanation for this is that the model of training in the UK at the time differed, with trainees more often operating without appropriate senior supervision.

Our findings are consistent with those of the New Zealand Joint Registry.[32 33] In a cohort of 35 415 THRs, of which 4049 were performed by trainees, the authors reported no significant difference in the revision rate between surgeon groups.[33] In a further cohort of 79 671 TKRs and 8854 UKRs, of which approximately 10% were performed by trainees, they reported no significant difference in the revision rates of knee replacements performed by trainees and consultants.[32] These studies were not included in this meta-analysis because the authors did not report net survival estimates and revision rates were reported 'per 100 component years', rather than for clearly defined periods of follow-up, which cannot be calculated from the data presented.

### Implications

There is a delicate balance between ensuring optimal outcomes for patients and the necessity to train the next generation of surgeons. Reidy *et al* suggest that the availability of surgeon level registry data as a means of benchmarking performance, may lead to a desire to avoid perceived poor performance and thus a reluctance among consultants to let trainees operate.[13 15] However, the findings of this review are encouraging and support the notion that in the context of contemporary practice, in countries with established and regulated orthopaedic training programmes, trainees can achieve implant survival outcomes equivalent to their consultant colleagues. The senior supervision of trainees was inconsistently reported in the studies included in this review but is likely to play an important role in the successful outcome of trainee performed hip and knee replacements.

An adequately powered non-inferiority randomised controlled trial (RCT) with 10 years follow-up assuming an acceptable revision rate of 5% and a 1% absolute non-inferiority delta ($\alpha$=0.05; power=0.80; 1:1 allocation ratio), would require a sample size of 6400 patients.[34] However, factors inherent to the training process, such as variation among trainees, the need for case selection according to complexity and varying levels of supervision based on a trainee's experience, may preclude an inclusive and therefore generalisable RCT. Further investigation should focus on the associations between senior supervision, specific surgeon training grade and the risk of revision following trainee-performed hip and knee replacements. Future work should also investigate the risk of early revision and the specific indications for revision following trainee-performed procedures. The analysis of unselected patient data recorded in a mandatory national joint replacement registry would be an appropriate means of further investigation.

## CONCLUSIONS

In conclusion, there is no strong evidence in the existing literature that trainee surgeons have worse outcomes than their consultant surgeon colleagues, in terms of the net survival, or crude revision rate of hip and knee replacements at 5–10 years follow-up. This may mean that there is no difference, or that appropriate case mix selection and supervision of trainees is currently employed and is safe to continue. Our results are concordant with published registry data,[32 33] and represent the best available evidence, but are limited by the quality of the existing published studies.

**Contributors** TJF, AWB, AS and MRW conceived and designed the study; TJF and ALA independently screened the articles and performed data extraction in duplicate; TJF and AS were responsible for data analysis; all authors were responsible for interpreting the data; TJF drafted the manuscript; AWB, ALA, AS and MRW revised the article critically for important intellectual content; all authors reviewed the final version of the manuscript and gave approval for submission for publication. The corresponding author attests that all listed authors meet authorship criteria and that no others meeting the criteria have been omitted. TJF is the guarantor.

**Funding** This study was supported by the National Institute for Health Research (NIHR) Bristol Biomedical Research Centre at the University Hospitals Bristol NHS Foundation Trust and the University of Bristol (grant number: N/A). TJF was supported by an NIHR Academic Clinical Fellowship. AS was supported by an MRC strategic skills fellowship (grant number: MR/L01226X/1).

**Disclaimer** The views expressed in this publication are those of the authors and not necessarily those of the NHS, the NIHR, or the Department of Health and Social Care. The NIHR had no role in the design and conduct of the study; the collection, management, analysis, and interpretation of the data; the preparation, review, or approval of the manuscript; or the decision to submit the manuscript for publication.

**Competing interests** AWB and MRW declare support from The Healthcare Quality Improvement Partnership/The NJR in the form of the Lot 2 contract for statistical analysis of the NJR, outside the submitted work; AWB and MRW report grants from the NIHR investigating the outcomes of joint replacement, outside the submitted work; AWB and MRW are editors of an Orthopaedic textbook for which they receive royalty payments from Taylor Francis; MRW reports fees paid to their institution for delivering teaching at courses organised by DePuy and Heraeus.

**Patient consent for publication** Not applicable.

**Provenance and peer review** Not commissioned; externally peer reviewed.

**Data availability statement** All data relevant to the study are included in the article or uploaded as online supplemental information.

**ORCID iDs**
Timothy J Fowler http://orcid.org/0000-0002-8195-0993
Adrian Sayers http://orcid.org/0000-0001-7452-5043
Michael R Whitehouse http://orcid.org/0000-0003-2436-9024

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
