## [Reviewer comments · BMJ Open]

ARTICLE DETAILS

TITLE (PROVISIONAL)	The association between surgeon grade and implant survival following hip and knee replacement: a systematic review and meta-analysis
AUTHORS	Fowler, Timothy; Aquilina, Alex; Blom, AW; Sayers, Adrian; Whitehouse, Michael

VERSION 1 – REVIEW

REVIEWER	A Pedersen Aarhus University
REVIEW RETURNED	08-Feb-2021

GENERAL COMMENTS	Abstract: 1. Conclusion should be based on the data presented in the result section. F.eks. only KM estimates are presented, but conclusion is made on cumulative incidence estimates too. Methods: 2. Please explain why the minimum follow up of five years was chosen. It is very likely that revisions due to infection, dislocation or fracture could occur more in trainee than in consultant group. 3. Why exclude studies reporting revision rate per 100 component years? The median follow up time is usually reported in these studies. 4. Please explain why studies that included pt. from 1990 were included. One can argue that diagnostic, treatment and training of surgeons has changed since 1990, and what is relevant in 1990 might not be relevant in 2010 or 2020. 5. The authors have nicely described primary exposure. It is however not clear from the review how the included studies have defined exposure, and if it is correlated to what authors have expected. It is very likely that the authors do not find any association due to misclassification of exposure. 6. Why only focus on KM and cumulative incidence measures of outcome? Discussion: 7. The authors state that risk of bias is moderate to severe, and that the quality of evidence for each outcome is low-very low. Please elaborate on these issues more in limitation section. 8. Could you describe the training program in UK during the last 30 years and changes of the same? I am just wondering if the “no association” found in this review is time-dependent.
---

REVIEWER	Richard de Steiger Epworth HealthCare, Department of Surgery
REVIEW RETURNED	09-Feb-2021

GENERAL COMMENTS	QUESTION TO EDITORS. The literature search was up till 10th December 2020. Was there enough time to contact authors, analyse data, write paper and have it for me to review by January 21st 2021. Extra ordinary! COMMENTS TO AUTHORS This is an excellent paper and gives further evidence for countries that have established orthopaedic training programs that, at least for hip and knee arthroplasty, the practice of supervised training does not result in adverse outcomes for patients. A few minor queries only. ABSTRACT - good INTRODUCTION Well set out introduction METHODS Exposure of interest clearly documented. Is there a reference for Rayyan or perhaps explain in a supplement? Very clear documentation of methods and how potential bias was addressed. RESULTS Although a large number of studies were found in the initial search, only a total of 5 studies for THR and 2 for TKR were eventually included. In the methods (Screening and data extraction) implant type is mentioned. Was there any analysis of implant type? I ask this question because it is unlikely there were enough numbers to do this but it would be interesting. With the AOANJRR we performed an analysis comparing private vs public (easier to do in Australia rather than UK). This could be a surrogate for consultant vs trainee. We found higher rates of revision in the private system but when adjusted for prosthesis type there was no difference. Perhaps a comment in discussion would be valuable. DISCUSSION I think at least some minimal discussion about the level of supervision a trainee may have might be appropriate. Although this was not part of the analysis, it is very briefly touched on in discussion. For example; trainee, consultant not in theatre, not scrubbed, or scrubbed and assisting are the levels of 'supervision' that trainees fill in their log books in Australia. Although there is unlikely to be information on this it is probably worth mentioning senior trainees from junior. Limitation section briefly mentions other outcome measures not reported. Perhaps dislocation (not revised) maybe higher in trainees and worth a comment about this.
---

REVIEWER	Jingheng Cai Sun Yat Sen Univ, Statistics
REVIEW RETURNED	16-Mar-2021

GENERAL COMMENTS	The manuscript employed meta analysis with K-M analysis to investigate the association between the training status of the surgeon and implant survival following primary hip and knee replacement. Given that I am not an expert in the medical field, I can only comment on the statistical analysis which is fine in this paper.
--

REVIEWER	Germain HONVO University of Liege, Division of Public Health, Epidemiology and Health Economics
REVIEW RETURNED	08-Apr-2021

GENERAL COMMENTS	The authors undertook a research to assess the association between surgeon training status and implant survival following hip and knee replacement through a systematic review and meta-analysis of observational studies. The rationale for the study has been clearly explained in the introduction. This research is welcome, as it helps better understanding whether survival of implants depends upon the training status of the surgeons. Here are some review comments that could help improving this manuscript.  1. In the introduction, line 40, a word (perhaps “compared”?) seems missing in the sentence “Radiographic studies have indicated that trainees achieve different implant alignment to their senior colleagues, ...”. Please check and revise. 2. Please consider moving the title “Data sources and search strategy” after the first paragraph of the methods section. 3. The authors searched only two bibliographic databases (Medline and Embase), instead of at least three databases which is the recommended strategy. Is there any reason for that choice? Otherwise, this may be reported as a limitation of this study in the discussion section. 4. Is there any particular reason for having performed separate literature searches for hip and knee replacement studies? Please explain this in the methods section. 5. Please change the title “quality assessment” to “Risk of bias and quality of evidence assessments”, to be in agreement with what is reported in this section. 6. The authors reported in the introduction that, as found in previous studies, “when trainees are appropriately supervised, they can obtain comparable functional outcomes and implant survivorship to their consultant colleagues...”. Therefore, why has this not been explored in the current study, by stratifying the analyses for outcomes in supervised versus non- supervised trainees? Perhaps because of the limited number of studies included? If so, this should be discussed, but first planned in the methods section. 7. Why have the authors chosen not to take into account potential confounding factors in individual studies in this meta-analysis? This needs to be explained in the methods section and discussed in the discussion section. This may constitute a limitation of this study. 8. Please, consider clearly distinguishing between assessment of quality of evidence (“quality assessment”) and assessment of risk of bias in individual studies. This comment applies for all sections of the manuscript. Please consider reporting the results for GRADE assessment (assessment of quality of evidence) after the results of
---

	the meta-analyses. On the contrary, the results of risk of bias in individual studies should be reported before. 9. Please consider significantly improving the reporting of the statistical analyses section, clearly distinguishing overall analyses from subgroup analyses for heterogeneity assessment. Variables for subgroup analyses should also be clearly reported. The sentence “We pooled survival estimates, assuming that survivorship approximated risk, with fixed effects meta-analysis weighting each study on the overall pooled estimate according to its standard error, which was calculated from published CIs;” is quite confusing. 10. As this is a meta-analysis of observational studies, there is no need to state in the results section that no randomized controlled trial was identified (Page 8, line 13). 11. Please avoid reporting simultaneously the 95% CI and p values. Also, there is no need to report the value of the tests performed (Z test for example). 12. Please consider reporting the I² values, whenever effects sizes with 95% CI are reported. When comparing subgroup effects sizes, I² values for subgroup differences (with corresponding p values) need to be reported as well.
--	---

VERSION 1 – AUTHOR RESPONSE

Reviewer: 1

Dr A Pedersen, Aarhus University

Response:

Dear Dr Pedersen, many thanks for your thorough and thoughtful review of our paper. We have tried to incorporate your suggestions where possible and hope to have adequately responded to your individual comments.

Comments to the Author:

Abstract:

1. Conclusion should be based on the data presented in the result section. F.eks. only KM estimates are presented, but conclusion is made on cumulative incidence estimates too.

Response:

We have now addressed your comment by amending the abstract to include the results for secondary outcome measure.

Methods:

2. Please explain why the minimum follow up of five years was chosen. It is very likely that revisions due to infection, dislocation or fracture could occur more in trainee than in consultant group.

Response:

Five-year intervals of follow-up were chosen for this study as they are commonly reported. Unfortunately, only a small proportion of observational studies on the subject publish lifetables that

enable earlier follow-up data to be extracted. We agree that earlier follow-up, e.g. at 1 year would be interesting to see and might capture early failures in the trainee group. Unfortunately, it was not possible in this study due to the limited available published data. We comment on this in our discussion and state that the five and ten-year intervals of follow-up are a limitation of this study, e.g.

“Meta-analysis of the primary outcome measure was only possible at five and ten years for THRs and ten years for TKRs, which limits the generalisability of our findings to these short and medium-term intervals of follow up.”

3. Why exclude studies reporting revision rate per 100 component years? The median follow-up time is usually reported in these studies.

Response:

With regards to the two NZJR papers in question (Inglis et al. & Storey et al.), median follow-up is not reported. We acknowledge that these two studies are key papers on the subject, thus we discuss the importance of their findings in our discussion and make the following statement on why they were not included:

“These studies were not included in this meta-analysis because the authors did not report net survival estimates and revision rates were reported ‘per 100 component years’, rather than for clearly defined periods of follow up, which cannot be calculated from the data presented.”

4. Please explain why studies that included pt. from 1990 were included. One can argue that diagnostic, treatment and training of surgeons has changed since 1990, and what is relevant in 1990 might not be relevant in 2010 or 2020.

Response:

Thank you for this comment, you make a very reasonable point, which we also considered carefully. We included studies from the past 30 years in an attempt to capture as many studies as possible, with as long follow-up as possible. It is a difficult balance in the context of training, where, as you say, practices are likely to have changed in that time. If we excluded studies with cases performed prior to 2010, then no published studies would meet the inclusion criteria. We comment on our justification for this in the methods section and make further comment on this in the limitations section.

5. The authors have nicely described primary exposure. It is however not clear from the review how the included studies have defined exposure, and if it is correlated to what authors have expected. It is very likely that the authors do not find any association due to misclassification of exposure.

Response:

The definition of the exposure in the included studies is summarised in Table 1 (6th column in Table 1). We have amended the title of this column to make this clearer to the reader. We are happy with the classification of exposure in the included studies and do not feel that this is a major limitation of the study.

6. Why only focus on KM and cumulative incidence measures of outcome?

Response:

Thank you for your comment. The aim of our study was, “To investigate the association between the training status of the surgeon and implant survival following primary hip and knee replacement.”

This study is part of a larger series of work looking at the role of training on implant survival and the longevity of hip and knee replacements. Therefore, we were only interested in implant survival outcomes (i.e. absence of failure) and did not look for, or collect data on functional outcomes (e.g. PROMS data). You may find it interesting to hear that no studies used competing risk models.

We make the following relevant comment in the limitations section of our discussion:

“Implant survival is a key determinant of good outcome in joint replacement surgery and is the sole variable considered in the current benchmarking strategies for the assessment of implant components. However, this review did not consider other factors that may be important when evaluating surgical outcomes, such as patient reported outcome measures, or complications other than failure.”

We agree that it would be interesting to look at additional outcome measures, but that was not the aim of this study.

Discussion:

7. The authors state that risk of bias is moderate to severe, and that the quality of evidence for each outcome is low-very low. Please elaborate on these issues more in limitation section.

Response:

A moderate to severe risk of bias (as per the ROBINS-I assessment method) and a low-very low quality of evidence (as per the GRADE approach for assessing quality) is consistent with the retrospective observational design of the small case series/cohort studies included in this study. The results of our GRADE and ROBINS-I assessments are reported in Table 2 and in Supplementary Table 3.

We clearly discuss these outcomes of these assessments in the limitations section where we state the following: “The GRADE assessment, which incorporates our risk of bias analysis, indicates the quality of evidence for each outcome to be low/very low, which is consistent with the predominantly retrospective design of included studies. Thus, the conclusions of this review are limited by the strength of the existing published data from a relatively small number of observational studies.”

Of note, there are no existing RCTs, or prospective clinical trials on this subject. At present, the studies included and discussed in this review, represent the best available data on the subject. However, we hope that we have made it clear that the existing evidence is of low quality, and that this is a limitation of the conclusions of our review. We have revised this part of the limitations section in response to your comments.

8. Could you describe the training program in UK during the last 30 years and changes of the same? I am just wondering if the “no association” found in this review is time-dependent.

Response:

Our review includes studies from the UK, France, Japan and Switzerland. While the majority of work on this subject has been conducted in the UK, we do not feel that this specific paper is the best place to include a detailed description of the history and format of UK Orthopaedic training programmes over the past 30 years, especially given the international origins of the included studies in this review.

Nonetheless, we agree that orthopaedic training practices are likely to have changed over the past 30 years and that this might influence the short to medium term survivorship of hip and knee replacements. The most significant way that orthopaedic training is likely to have changed in the past three decades is that trainees today are probably more likely to be supervised by a consultant than they were 30 years ago. We have anecdotal experience to suggest this. However, as we discuss in the limitations of our study (and displayed in Table 1), the level of supervision of trainees was only reported in two of the eight included studies. Therefore, it is not feasible to quantify, or adjust for the level of supervision of trainees in this study.

Importantly, our findings were consistent with the two New Zealand Joint Registry studies that we discuss in the paper. Inglis et al. included patients from 2005-2012; Storey et al included patients from 1999-2016, both shorter periods than this systematic review. Further registry work on this subject is warranted. We have amended the limitations section accordingly.

Reviewer: 2
Dr Richard de Steiger

Response:

Dear Dr de Steiger, many thanks for the time you have put into reviewing our article and for the insight that you provide. We hope that we have appropriately addressed your comments.

Comments to the Author:

QUESTION TO EDITORS. The literature search was up till 10th December 2020. Was there enough time to contact authors, analyse data, write paper and have it for me to review by January 21st 2021. Extra ordinary!

Response:

This study was originally conducted in early 2020 and a significant amount of time was spent communicating with corresponding authors and preparing the manuscript. The paper spent a number of months with another BMJ journal, before being resubmitted to BMJ Open in December 2020. The search was repeated prior to this submission, which confirmed that no additional studies had been published on the subject/met the inclusion criteria. We hope this explains the seemingly quick turnaround.

COMMENTS TO AUTHORS

This is an excellent paper and gives further evidence for countries that have established orthopaedic training programs that, at least for hip and knee arthroplasty, the practice of supervised training does not result in adverse outcomes for patients. A few minor queries only.

ABSTRACT - good

INTRODUCTION

Well set out introduction

METHODS

Exposure of interest clearly documented. Is there a reference for Rayyan or perhaps explain in a supplement? Very clear documentation of methods and how potential bias was addressed.

Response:

We have now added a reference for Rayyan.

RESULTS

Although a large number of studies were found in the initial search, only a total of 5 studies for THR and 2 for TKR were eventually included. In the methods (Screening and data extraction) implant type is mentioned. Was there any analysis of implant type? I ask this question because it is unlikely there were enough numbers to do this, but it would be interesting. With the AOANJRR we performed an analysis comparing private vs public (easier to do in Australia rather than UK). This could be a surrogate for consultant vs trainee. We found higher rates of revision in the private system but when adjusted for prosthesis type there was no difference. Perhaps a comment in discussion would be valuable.

Response:

Unfortunately, there was no analysis/adjustment according to implant design (including fixation, bearing surface, or head size, etc). We make the following comment in our limitations section: "Published literature did not consistently report age, sex, comorbidities, implant design, or the level of senior supervision; making it very difficult to adjust for these variables."

We agree that it would be optimal and interesting to adjust for measures of implant design (including fixation, head size and bearing surface). It was not feasible in this review, but we are doing this in further studies.

DISCUSSION

I think at least some minimal discussion about the level of supervision a trainee may have might be appropriate. Although this was not part of the analysis, it is very briefly touched on in discussion. For example; trainee, consultant not in theatre, not scrubbed, or scrubbed and assisting are the levels of 'supervision' that trainees fill in their log books in Australia. Although there is unlikely to be information on this it is probably worth mentioning senior trainees from junior.

Response:

Thank you for this comment. We completely agree and have amended the limitations section to further emphasise the importance of variations in the level of experience between trainees (which is not captured using a binary variable such as 'training status'), and supervision. Unfortunately, the included studies did not consistently report the specific training grade, or level of supervision of trainees (only two papers reported the level of supervision). We have now expanded on this further in the limitations section.

Similar information on the specific level of supervision (e.g. scrubbed, or not scrubbed) is captured in UK training logbooks and in the National Joint Registry. While we are unable to comprehensively address these matters in this study (due to limitations in the available data), this is a focus of our ongoing work on the subject.

Limitation section briefly mentions other outcome measures not reported. Perhaps dislocation (not revised) maybe higher in trainees and worth a comment about this.

Response:

We agree with this comment and have amended the discussion accordingly.

Reviewer: 3

Dr Jingheng Cai, Sun Yat Sen Univ

Comments to the Author:

The manuscript employed meta-analysis with K-M analysis to investigate the association between the training status of the surgeon and implant survival following primary hip and knee replacement.

Given that I am not an expert in the medical field, I can only comment on the statistical analysis which is fine in this paper.

Response:

Dear Dr Cai, thank you for the time that you have taken to review our paper. Thank you for your comment and your approval of the statistical methods used in our paper.

Reviewer: 4

Dr Germain HONVO, University of Liege, University of Abomey-Calavi

Comments to the Author:

The authors undertook a research to assess the association between surgeon training status and implant survival following hip and knee replacement through a systematic review and meta-analysis of observational studies. The rationale for the study has been clearly explained in the introduction. This research is welcome, as it helps better understanding whether survival of implants depends upon the training status of the surgeons. Here are some review comments that could help improving this manuscript.

Response:

Dear Dr Honvo, thank you for your very detailed review and comments. We are grateful for the time that you have taken to thoroughly review our manuscript and the insight that you provide. We hope that we have been able to address your comments to a satisfactory standard.

1. In the introduction, line 40, a word (perhaps “compared”?) seems missing in the sentence “Radiographic studies have indicated that trainees achieve different implant alignment to their senior colleagues, ...”. Please check and revise.

Response:

We have checked the sentence structure and revised this accordingly.

2. Please consider moving the title “Data sources and search strategy” after the first paragraph of the methods section.

Response:

We have moved this subtitle according to your suggestion.

3. The authors searched only two bibliographic databases (Medline and Embase), instead of at least three databases which is the recommended strategy. Is there any reason for that choice? Otherwise, this may be reported as a limitation of this study in the discussion section.

Response:

It is our understanding that a search of two bibliographic libraries is generally acceptable. Neither the PRISMA, nor MOOSE checklists specify a minimum number. Our group published the following two studies in The Lancet in 2019 (PMID: 30782341; PMID: 30782340), both with searches of just MEDLINE and Embase.

4. Is there any particular reason for having performed separate literature searches for hip and knee replacement studies? Please explain this in the methods section.

Response:

Separate searches were used in order to emphasise the inherent differences between hip and knee replacements as distinct treatments and in an attempt to make the search process clearer and more detailed for readers. Separate studies could have feasibly been conducted for each intervention but given the paucity of existing work on this subject (only 8 studies in total) it seems reasonable to combine both hip and knee replacements into a single study as we have done here. We acknowledge that we have displayed the flow diagram in Figure 1 in a novel manner. We feel that this emphasises the important and inherent differences in the interventions.

5. Please change the title “quality assessment” to “Risk of bias and quality of evidence assessments”, to be in agreement with what is reported in this section.

Response:

Thank you for your suggestion. Change made as recommended.

6. The authors reported in the introduction that, as found in previous studies, “when trainees are appropriately supervised, they can obtain comparable functional outcomes and implant survivorship to their consultant colleagues...”. Therefore, why has this not been explored in the current study, by stratifying the analyses for outcomes in supervised versus non-supervised trainees? Perhaps because of the limited number of studies included? If so, this should be discussed, but first planned in the methods section.

Response:

The aim of our study, as documented in the first line of the abstract was: “To investigate the association between the training status of the surgeon (i.e. trainee vs. consultant) and implant survival following primary hip and knee replacement.”

This study contributes to a larger body of work on the associations between surgical training and implant survival outcomes following hip and knee replacement. It was not our aim to explore functional outcomes as part of this study. However, we acknowledge that it would be interesting to see further work on this subject.

Table 1 shows that the supervision of trainees was only reported in two papers (Reidy – THR, and Bottomley – UKR). No other studies reported the supervision of trainees; thus, we are unable to stratify the analysis for outcomes according to supervision. We have edited the discussion to make this limitation clear to the reader.

The role of supervision is the focus of our ongoing work on this subject. We have added a comment on ‘further investigation’ to the discussion.

7. Why have the authors chosen not to take into account potential confounding factors in individual studies in this meta-analysis? This needs to be explained in the methods section and discussed in the discussion section. This may constitute a limitation of this study.

Response:

Potential confounding factors including age, sex, implant design, indication and supervision are reported in Table 1. Due to the small number of studies and inconsistent reporting it would be very difficult to meaningfully adjust for these potential confounding factors. We have discussed this as a limitation of the study.

8. Please, consider clearly distinguishing between assessment of quality of evidence (“quality assessment”) and assessment of risk of bias in individual studies. This comment applies for all sections of the manuscript. Please consider reporting the results for GRADE assessment (assessment of quality of evidence) after the results of the meta-analyses. On the contrary, the results of risk of bias in individual studies should be reported before.

Response:

We appreciate your insight and experience on this subject and have made the relevant changes according to your recommendations.

9. Please consider significantly improving the reporting of the statistical analyses section, clearly distinguishing overall analyses from subgroup analyses for heterogeneity assessment. Variables for subgroup analyses should also be clearly reported. The sentence “We pooled survival estimates, assuming that survivorship approximated risk, with fixed effects meta-analysis weighting each study on the overall pooled estimate according to its standard error, which was calculated from published CIs;” is quite confusing.

Response:

Thank you for your comment, we have considered this very carefully. With regards to our statistical methods and the following sentence, “We pooled survival estimates, assuming that survivorship approximated risk, with fixed effects meta-analysis weighting each study on the overall pooled estimate according to its standard error, which was calculated from published CIs.” This is an established statistical method that our group has previously published on a number of occasions - twice in The Lancet (PMID: 30782341; PMID: 30782340), and again in The Lancet Rheumatology ([https://doi.org/10.1016/S2665-9913\(20\)30226-5](https://doi.org/10.1016/S2665-9913(20)30226-5)). We have been careful to use and describe this method in similar terms to these previous studies and have cited them in order to draw attention the

prior use of this statistical method, which has been peer-reviewed on a number of occasions. We feel that the description of the statistical methods in our paper is accurate and pitched at an appropriate level for our intended readers. Further subgroup analysis was not performed and so has not been described.

10. As this is a meta-analysis of observational studies, there is no need to state in the results section that no randomized controlled trial was identified (Page 8, line 13).

Response:

This comment has now been deleted.

11. Please avoid reporting simultaneously the 95% CI and p values. Also, there is no need to report the value of the tests performed (Z test for example).

Response:

Thank you for this comment. We have now deleted test values (including Z test). We have now made the recommended edits in order to avoid the simultaneous reporting of p values and confidence intervals.

12. Please consider reporting the I² values, whenever effects sizes with 95% CI are reported. When comparing subgroup effects sizes, I² values for subgroup differences (with corresponding p values) need to be reported as well.

Response:

The I² values are reported (along with corresponding 95% CIs and p-values where appropriate) in Figures 2-3.

VERSION 2 – REVIEW

REVIEWER	A Pedersen Aarhus University
REVIEW RETURNED	01-Jun-2021

GENERAL COMMENTS	Abstract:  1. results: It seems somewhat misleading that the authors write that 8 studies are included. The results section should highlight the fact that data synthesis was done on five (THR) or two (TKR) studies and was not possible for some outcomes, due to sparse data (net implant survival 10 years (THR) and 5 years (TKR), crude revision rate (TKR) and all outcomes (UKR)). 2. conclusion: The authors should mention that few studies and high risk of bias/low or very low quality of included studies makes the conclusion uncertain: ie “no strong evidence” wording can be misleading. Please rephrase the conclusion. Introduction:  1. Hypothesis statement is lacking. 2. The sentence “However, the causative impact of these findings....” should be deleted. Studies 7-16 are observational and
--

	causality cannot be established, not even in the current study. 3. Can authors elaborate on WHY patients operated by trainees suffer more complications such as operation time, dislocation, major systemic complications (pulmonary embolism etc), and radiological differences, but less revisions (References Hedlundh et al. (7) and Schoenfeld et al. (8), Moran et al (9), Kim et al (10) and Kazarian et al (11)). If the low revision rate (And RR in favor of trainee) is explained by appropriate patient selection and supervision by a senior surgeon, then why does it only apply to implant survival. Methods:  1. Suggest to use PICO approach to pose research question and objectives 2. Eligibility criteria, as well as inclusion and exclusion criteria should be stated more clearly in method section, and it should be clear which criteria has been applied during title/abstract screening, and which during full-text reading. Flow diagram shows that 1076 hip and 564 knee studies were doomed as irrelevant. However, it is not clear what criteria are used to make that decision. 3. The authors have explained in review response that only five and 10 years follow up analyses were possible. Please include in the discussion section some thoughts about the impact of revisions due to infection, dislocation or fracture that might occur more in trainee than in consultant group during the 1-2 years after primary procedure and could dilute any potential association. 4. Rationale for using fixed effects meta-analyses rather than random effect model should be provided. Discussion:  1. The authors here touch upon a very important confounder: experienced surgeons/consultants might be selected for the difficult cases with a much higher risk of revision. I think this matter could be elaborated a bit: have any of the included studies made attempts to adjust for this confounder in any way and what are the views on this matter from other studies – perhaps from other fields of surgery if none exists in this particular field. 2. On page 11, in the discussion section, line 32-39 the author write: “ The GRADE assessment indicates a low to very low quality of evidence for each outcome. Furthermore, the ROBINS-I assessment indicates a moderate to severe risk of bias in the included studies. These findings are consistent with the predominantly retrospective design of the included studies.” Does this statement means that retrospective design automatically equals risk of bias? Please clarify. Conclusion:  1. Please see the previous comment regarding wording “no strong evidence” and rephrase the conclusion. Two phd students from the Department of Clinical Epidemiology, Aarhus University have been of assistance for this review: Nadia Roldsgaard Gadgaard, phd student Thomas Johannesson Hjelholt, phd student.
--	---

REVIEWER	Richard de Steiger Epworth HealthCare, Department of Surgery
REVIEW RETURNED	28-May-2021

GENERAL COMMENTS	Dear Author & Editor Happy the majority of queries were answered. For future reference, normally when a reviewer reads the authors comments, the author sets out Reviewer 1 and replies to comments point by point and then states where this was addressed in the manuscript (if needed). I did not see this done here but just comments on manuscript which made it hard to review again.
--

REVIEWER	Jingheng Cai Sun Yat Sen Univ, Statistics
REVIEW RETURNED	20-May-2021

GENERAL COMMENTS	I have no further comments.
-----------------------------

REVIEWER	Germain HONVO University of Liege, Division of Public Health, Epidemiology and Health Economics
REVIEW RETURNED	31-May-2021

GENERAL COMMENTS	The reviewer thanks the authors for their careful consideration of comments and suggestions made during the first peer review round. Here are some final few comments. In response to previous comment on the fact that the authors searched only two bibliographic databases, instead of at least three, the authors answered that “neither the PRISMA, nor MOOSE checklists specify a minimum number”. Though having searched at least Medline and Embase is acceptable, the reviewer would like to draw the authors’ attention on the fact that the more databases are considered for literature search, the more likely almost all the available studies on a specific topic are captured. For future systematic reviews, the reviewer strongly suggests to the authors to search at least three relevant databases, even if not specifically recommended in the PRISMA guideline. More generally, the reviewer would like to kindly draw the authors’ attention on the fact that being published in the Lancet is not a warrant of meeting the best standards or of being of good quality. The reviewer was unable to find any difference in Figure 1, in the current version of the manuscript, compared to the first version. Please check and correct, if necessary.
--

VERSION 2 – AUTHOR RESPONSE

Reviewer: 1

Dr A Pedersen, Aarhus University

Response:

Dear Dr Pedersen, thank you again for your review of our paper. Please also extend our thanks to Nadia Roldsgaard Gadgaard and Thomas Johannesson Hjelholt for their assistance with this review.

We feel that the edits that we have made in response have enhanced this paper. We have tried to incorporate your suggestions where possible and hope to have adequately responded to your individual comments.

Comments to the Author:

Abstract:

1. It seems somewhat misleading that the authors write that 8 studies are included. The results section should highlight the fact that data synthesis was done on five (THR) or two (TKR) studies and was not possible for some outcomes, due to sparse data (net implant survival 10 years (THR) and 5 years (TKR), crude revision rate (TKR) and all outcomes (UKR)).

Response:

Thank you for your comment. We have now clarified this point in the abstract results section, with the addition of the following statement “(five THR studies, two TKR studies, and two UKR studies)”. This takes the abstract up to the maximum limit of 300 words. We agree that it is important to discuss that data synthesis was only possible for some outcomes due to sparse data and we discuss this in detail in the main manuscript (please see our results section and detailed discussion of limitations). However, unfortunately we are unable to include this in the abstract without deleting other mandatory information, or significantly exceeding the word limit. We hope that you find our revision a satisfactory compromise.

2. Conclusion: The authors should mention that few studies and high risk of bias/low or very low quality of included studies makes the conclusion uncertain: i.e., “no strong evidence” wording can be misleading. Please rephrase the conclusion.

Response:

We agree with your comment. We agree that the strengths of the conclusions from this paper are limited by the strength and quality of the existing published data, this is already detailed in our discussion. As with our above statement, we are unfortunately limited by the word limit, thus cannot add a detailed discussion of the paper’s limitations to the abstract. However, we have added a short statement to acknowledge the limitations of this study. The final sentence now reads: “*These findings are limited by the strength and quality of the existing published data and are applicable to countries with established orthopaedic training programmes.*”

Introduction:

1. Hypothesis statement is lacking.

Response:

It is our understanding that a systematic review/meta-analysis should state a hypothesis OR state a specific question. It is our preference to specifically state the research question. We have added a sentence to the end of the introduction to make this clearer to the reader. *“We aimed to answer the question – do trainees achieve equivalent implant survival outcomes to consultants when performing primary hip and knee replacement?”*

2. The sentence *“However, the causative impact of these findings...”* should be deleted. Studies 7-16 are observational, and causality cannot be established, not even in the current study.

Response:

We agree that causation cannot be attributed to the findings of an observational study. The sentence in question (*“However, the causative impact of these findings on implant survival has not been established.”*) does not contradict this statement. However, to avoid any misunderstanding, we have edited this sentence to read as follows: *“However, the relative impact of these findings on implant survival has not been established”*.

3. Can authors elaborate on WHY patients operated by trainees suffer more complications such as operation time, dislocation, major systemic complications (pulmonary embolism etc), and radiological differences, but less revisions (References Hedlundh et al. (7) and Schoenfeld et al. (8), Moran et al (9), Kim et al (10) and Kazarian et al (11)). If the low revision rate (And RR in favor of trainee) is explained by appropriate patient selection and supervision by a senior surgeon, then why does it only apply to implant survival.

Response:

The paragraph in question outlines the breadth and summarises the principal findings in the existing published literature relating to outcomes following trainee participation in hip and knee replacement surgery. It was not the aim of our study to explain a link/association between the higher rates of complications and longer operative times that have been identified in orthopaedic procedures performed by trainees. The aim of our study was to perform evidence synthesis *“on the association between surgeon grade (trainee vs. consultant) and implant survival outcomes in hip and knee replacement surgery.”* Therefore, while the question you pose is interesting, it is outside the defined scope of this study to explain the causative link between complications, operative duration, and the risk of revision.

Methods:

1. Suggest to use PICO approach to pose research question and objectives

Response:

We would like to assure you that we have used a PICO approach through the design and conduct of this study. It is our preference to write this in prose, as we have done in the manuscript, rather than as below:

Participants: Adult patients (≥ 18 years old) undergoing primary hip or knee replacement predominantly for the treatment of osteoarthritis (*defined under the heading 'Eligibility criteria'*).

Intervention: The surgeon recorded as performing the procedure was a trainee (*defined under the heading 'Primary exposure'*).

Comparison: The surgeon recorded as performing the procedure was a consultant (*defined under the heading 'Primary exposure'*).

Outcomes: The primary outcome was net implant survival, reported as a Kaplan-Meier survival estimate. The secondary outcome measure was crude revision rate, which was defined as the observed number of revisions in a specified period of time (*defined under the heading 'Outcome measures'*).

2. Eligibility criteria, as well as inclusion and exclusion criteria should be stated more clearly in method section, and it should be clear which criteria has been applied during title/abstract screening, and which during full-text reading. Flow diagram shows that 1076 hip and 564 knee studies were deemed irrelevant. However, it is not clear what criteria are used to make that decision.

Response:

Thank you for this feedback. We feel that the inclusion and exclusion criteria are adequately defined within the text. However, in response to your comment, we have added a section to the supplementary material, which sets out the inclusion and exclusion criteria in clear, concise bullet points. Studies were initially screened for inclusion according to information contained within the titles and abstracts, according to standard practice. We have added a sentence to clarify this in the 'screening and data extraction' section of the methods section. Specific reasons for exclusion are documented for studies that were excluded following full-text review. We have amended the 'screening and data extraction' section of the methods section to make this clearer to the reader. Furthermore, we have amended figure 1 to state this more clearly.

3. The authors have explained in review response that only five and 10 years follow up analyses were possible. Please include in the discussion section some thoughts about the impact of revisions due to infection, dislocation or fracture that might occur more in trainee than in consultant group during the 1-2 years after primary procedure and could dilute any potential association.

Response:

We completely agree. It would be very interesting to investigate: 1) the differences in early failure rates, and 2) the indications for revision between trainee and consultant-performed procedures. This is the focus of our ongoing work in a number of separate studies. These are questions that cannot be answered in this systematic review, as the answers are not yet present in published literature.

We have edited the following two paragraphs in the discussion to highlight this limitation of our study and to outline the need for further work on this subject:

- 1) Strengths and limitations section of the discussion: *“Meta-analysis of the primary outcome measure was only possible at five and ten years for THRs and ten years for TKRs, which limits the generalisability of our findings to these short and medium-term intervals of follow up. Thus, this review does not capture any differences in early failure rates that might exist between trainee and consultant cohorts before five years.”*
- 2) Paragraph on future work at the end of the discussion: *“Further investigation should focus on the associations between senior supervision, specific surgeon training grade, and the risk of revision following trainee-performed hip and knee replacements. Future work should also investigate the risk of early revision and the specific indications for revision following trainee-performed procedures. The analysis of unselected patient data recorded in a mandatory national joint replacement registry would be an appropriate means of further investigation.”*

4. Rationale for using fixed effects meta-analyses rather than random effect model should be provided.

Response:

There are two published examples of statistical methods used for the meta-analysis implant survival estimates; both use a fixed effects model. We have replicated and appropriately cited this published

method. The use of a fixed effects model reflects the common treatment method (e.g. THR) and the common, clearly defined outcome, implant failure.

“We pooled survival estimates, assuming that survivorship approximated risk, with fixed effects meta-analysis weighting each study on the overall pooled estimate according to its standard error, which was calculated from published CIs; a method for the meta-analysis of implant survival estimates described by Evans et al.^{4,5}”

4. Evans JT, Evans JP, Walker RW, et al. How long does a hip replacement last? A systematic review and meta-analysis of case series and national registry reports with more than 15 years of follow-up. *Lancet*. 2019;**393**(10172):647-54.

5. Evans JT, Walker RW, Evans JP, et al. How long does a knee replacement last? A systematic review and meta-analysis of case series and national registry reports with more than 15 years of follow-up. *Lancet*. 2019;**393**(10172):655-63.

Discussion:

1. The authors here touch upon a very important confounder: experienced surgeons/consultants might be selected for the difficult cases with a much higher risk of revision. I think this matter could be elaborated a bit: have any of the included studies made attempts to adjust for this confounder in any way and what are the views on this matter from other studies – perhaps from other fields of surgery if none exists in this particular field.

Response:

Case complexity a complex and poorly defined concept. Clearly some cases are more complex than others, and in general, more experienced surgeons are likely to perform the more complex procedures. However, none of the included studies attempted to adjust for case complexity.

Methods of categorising case complexity in joint replacement surgery have been proposed (for example for revision knee replacement: <https://doi.org/10.1007/s00167-019-05462-x> – please note we have not cited this system, which is not applicable to primary knee replacement, and therefore not relevant to this study). However, uptake of these methods is sparse and inconsistently utilised in clinical practice, let alone research.

For example, the National Joint Registry for England and Wales (The NJR), which is the largest joint replacement registry in the world, does not use a formal method of categorising case complexity. Instead, studies using registry data tend to adjust for the numerous patient, operation and unit level factors that they record (using these as a surrogate for complexity).

Until we have a globally accepted method of categorising case complexity, adjusting for this confounder will not be possible in evidence synthesis. A detailed discussion of case complexity and this topic is not within the scope of this paper. However, we have attempted to highlight this issue more clearly in our discussion of the limitations.

“Published literature did not consistently report age, sex, comorbidities, implant design, or the level of senior supervision; making it very difficult to adjust for these variables. Methods of categorising the procedural complexity of a hip or knee replacement are not widely used in the orthopaedic literature and were not reported by any of the studies included in this review. Therefore, it was not possible to adjust for this factor. It is reasonable to suggest that the predominantly superior survival outcomes observed in the trainee cohorts are a product of patient selection and close senior supervision, with good trainers selecting appropriately complex cases for their trainees.”

2. On page 11, in the discussion section, line 32-39 the author write:

“ The GRADE assessment indicates a low to very low quality of evidence for each outcome. Furthermore, the ROBINS-I assessment indicates a moderate to severe risk of bias in the included studies. These findings are consistent with the predominantly retrospective design of the included studies.”

Does this statement means that retrospective design automatically equals risk of bias? Please clarify.

Response:

In general, studies with a retrospective design have a higher risk of bias than well-designed prospective studies, such as randomised controlled trials. This is not automatic. We acknowledge that retrospective studies can be very well designed and offer valuable contributions to the scientific literature. Our statement draws the reader’s attention to the fact that the studies included in this review had a predominantly retrospective design, which is generally consistent with their low quality and high risk of bias.

We have edited this sentence accordingly and it now reads as follows: *“These findings are generally consistent with the predominantly retrospective design of the included studies.”*

Conclusion:

1. Please see the previous comment regarding wording “no strong evidence” and rephrase the conclusion.

Thank you for your comment. We have amended the conclusions to reflect your previous comment and added a final sentence, which acknowledges the low quality of included studies.

“Our results are concordant with published registry data, and represent the best available evidence, but are limited by the quality of the existing published studies.”

We have thought very carefully about our use of the term “no strong evidence”, or “no strong evidence of an association”. We prefer this term to alternative such as “statistically significant” and feel that it reflects a cautious and realistic interpretation of the results, which are based on a relatively small number of observational studies. It would be our preference to leave this terminology unchanged.

Reviewer: 2

Dr Richard de Steiger

Response:

Dear Dr de Steiger,

Thank you again for the time you have taken to review our paper. Thanks for your comment, which we have taken on board. Please note that the journal’s submission software automatically applies page and line numbers making it difficult to reference these in this reply. We have added clear comments to the marked version of the manuscript to outline how we have responded to each reviewer’s queries and included any changes in this document in case you do not receive the marked version of the manuscript.

Comments to the Author:

Happy the majority of queries were answered.

For future reference, normally when a reviewer reads the authors comments, the author sets out Reviewer 1 and replies to comments point by point and then states where this was addressed in the manuscript (if needed). I did not see this done here but just comments on manuscript which made it hard to review again.

Reviewer: 3

Dr Jingheng Cai, Sun Yat Sen Univ

Comments to the Author:

I have no further comments.

Response:

Dear Dr Cai, thanks again for the time that you have taken to review our paper. Thank you for your comment and your approval of the statistical methods used in our paper.

Reviewer: 4

Dr Germain HONVO, University of Liege, University of Abomey-Calavi

Comments to the Author:

The reviewer thanks the authors for their careful consideration of comments and suggestions made during the first peer review round. Here are some final few comments.

In response to previous comment on the fact that the authors searched only two bibliographic databases, instead of at least three, the authors answered that “neither the PRISMA, nor MOOSE checklists specify a minimum number”. Though having searched at least Medline and Embase is acceptable, the reviewer would like to draw the authors’ attention on the fact that the more databases are considered for literature search, the more likely almost all the available studies on a specific topic are captured. For future systematic reviews, the reviewer strongly suggests to the authors to search at least three relevant databases, even if not specifically recommended in the PRISMA guideline.

More generally, the reviewer would like to kindly draw the authors’ attention on the fact that being published in the Lancet is not a warrant of meeting the best standards or of being of good quality.

The reviewer was unable to find any difference in Figure 1, in the current version of the manuscript, compared to the first version. Please check and correct, if necessary.

Response:

Dear Dr Honvo, thanks again for the time you have taken to review our manuscript. We appreciate your comments regarding bibliographic databases and will certainly keep this in mind for future studies.

With regards to Figure 1: given the time between submissions (10 months since initial submission), I have updated the search and updated Figure 1 accordingly. You will note from the cover letter, that this led to the inclusion of an additional study of UKRs. Cover letter statement as follows: *“In repeating the literature search, we have identified and included one additional study of unicompartmental knee replacement. The inclusion of this paper does not alter the interpretation of findings, or the*

conclusions of this study. The results section, discussion, figures, and tables have been cautiously updated to account for this minor amendment, whilst minimising any changes to the manuscript as a whole.”